# An Old Technique with A Promising Future: Recent Advances in the Use of Electrodeposition for Metal Recovery

**DOI:** 10.3390/molecules26185525

**Published:** 2021-09-11

**Authors:** Yelitza Delgado, Francisco J. Fernández-Morales, Javier Llanos

**Affiliations:** Chemical Engineering Department, University of Castilla-La Mancha, 13071 Ciudad Real, Spain; yelitza.delgado@uclm.es (Y.D.); fcojesus.fmorales@uclm.es (F.J.F.-M.)

**Keywords:** electrodeposition, metal recovery, ionic liquids, bioelectrochemistry, integration of technologies

## Abstract

Although the first published works on electrodeposition dates from more than one century ago (1905), the uses of this technique in the recovery of metals are attracting an increasing interest from the scientific community in the recent years. Moreover, the intense use of metals in electronics and the necessity to assure a second life of these devices in a context of circular economy, have increased the interest of the scientific community on electrodeposition, with almost 3000 works published per year nowadays. In this review, we aim to revise the most relevant and recent publications in the application of electrodeposition for metal recovery. These contributions have been classified into four main groups of approaches: (1) treatment and reuse of wastewater; (2) use of ionic liquids; (3) use of bio-electrochemical processes (microbial fuel cells and microbial electrolysis cells) and (4) integration of electrodeposition with other processes (bioleaching, adsorption, membrane processes, etc.). This would increase the awareness about the importance of the technology and would serve as a starting point for anyone that aims to start working in the field.

## 1. Introduction

Electrodeposition (ED) has been studied as one of the key electrochemical technologies for more than a century. According to the Scopus database, the first article about electrodeposition was published in 1905 [1] and studied the electrodeposition of copper upon iron. Since then, the scientific community has added new materials, methods, conditions and many other elements in order to improve this technique and its results and to widen the range of potential applications. This growing interest has caused an exponential increase in the number of publications regarding electrodeposition. Thus, Figure 1 shows the number of works published, according to the Scopus database, related to the terms “electrodeposition” or “electrochemical deposition”.

As it can be seen in Figure 1, there is an exponential increase in the number of articles published in this topic, reaching a total amount of almost 56,000 papers and 3000 articles in the recent 2020 (1771 in 2021 until 23 July). This evolution means that, although electrodeposition is a mature technology, the range of applications in which this technique can be applied is widening, increasing the interest of the scientific community on its development.

In order to focus on the topic of this review, the term “metal recovery” was included to refine this search. The results of this refined search in the Scopus database were analyzed by the software *VOSviewer*, a free software developed by Nees Jan van Eck and Ludo Waltman at Leiden University’s Centre for Science and Technology Studies (CWTS). Two figures were created in order to show the countries that have devoted more interest to this topic (Figure 2) and the most used keywords (Figure 3). Moreover, the connections between papers are also shown in these figures. In the case of Figure 2, it was restricted to countries with a minimum of 13 publications, meanwhile for Figure 3, a minimum number of occurrences of an author keyword of 13 was also used in order to limit the complexity of the figures.

Almost 1300 papers were found for this topic. As it can be observed in Figure 2, China appears as the country which has devoted the highest interest in the development of electrodeposition for metal recovery. It is followed by the United States, United Kingdom, India and Japan. Regarding Figure 3, apart from electrodeposition and metal recovery (terms included in the search restrictions), recycling, leaching, different metals, ionic liquids, microbial electrolysis cell, microbial fuel cell and wastewater treatment are among the most common topics in this research field.

The use of electrodeposition for the recovery of metal ions is a topic closely related to environmental applications, as the presence of heavy metals represents an environmental issue of increasing concern, due to the massive use of metals in electronics and other applications and its inherent bioaccumulation and potential risks for the human health [2,3,4,5,6]. If this treatment is performed by electrodeposition, the removal of heavy metals is performed together with its recovery, working on the field of circular economy, an essential part of the main research programs worldwide, including Horizon Europe of the European Commission.

Based on this, the main aim of this paper is to review the most recent and relevant applications of electrodeposition, focusing on the recovery of metals. According to the most common topics of recent research work, the articles have been classified into four different groups: (1) treatment and reuse of wastewater; (2) use of ionic liquids; (3) use of bio-electrochemical processes (microbial fuel cells and microbial electrolysis cells) and (4) integration of electrodeposition with other processes (bioleaching, adsorption, electrodialysis, etc.).

## 2. Treatment and Reuse of Wastewater by Electrodeposition

Wastewater from industries, hospitals and domestic applications are an environmental issue of increasing concern nowadays [7,8,9,10]. As it is stated in several of the United Nations Global Development Goals and it all the research programs worldwide, it is compulsory to look for water treatments in order to obtain clean water without metals, plastics and microplastics, pesticides and other pollutants [11,12,13].

In this field, electrodeposition serves as a plausible technology to recover a wide spectrum of metals and to remove them from the target effluents to be treated. Table 1 gathers some of the most recent works in this field, including the main objective of the work, the effluent treated and the main conclusions obtained.

Generally, the articles published in the field are divided between those treating real wastewater [14,15,16,17] and those that elaborate a synthetic one [18,19,20,21,22] in order to simulate real effluents to set the material required for the installation, operational conditions, parameters to measure, elements to quantify and the equipment to use for it.

The common presence of copper in both electronic or deplating wastewaters joined to its high value of reduction potential (+0.34 V vs. SHE) and its consequent facile electrodeposition. This metal is the most commonly studied in papers regarding the recovery of metals by electrodeposition, although the recovery of other metals such as Co, Ni, Pd, Pb, Zn or Te is also evaluated, as can be observed in Table 1.

In the recent papers published about metal recovery by electrodeposition, it is possible to find several approaches to enhance the overall behavior of the process. The first interesting approach is coupling the cathodic recovery of target metals to the anodic oxidation of organic matter, as it is the case of the work of Gu et al. [14], who studied the reduction of chemical oxygen demand (COD) by oxidation of plastic deplating wastewater, obtaining a reduction of COD from 1360 mg L^−1^ to 378 mg L^−1^ after the electrochemical oxidation and a subsequent oxidation by H_2_O_2_. In the same line, Gu et al. [16] evaluated the simultaneous removal of phenol and recovery of several heavy metals from petrochemical wastewaters.


molecules-26-05525-t001_Table 1Table 1Most recent articles published for treatment and reuse of wastewater (WW) by electrodeposition for metal recovery.ObjectiveWater Composition/Process ConditionsConclusionsRef.Remove Ag^+^ by precipitation, Cu^2+^ by electrodeposition and reduce COD by oxidation of real wastewater from plastic deplating.WW Composition: 5600 mg·L^−1^ Ag^+^; 9069 mg·L^−1^ Cu^2+^; 1360.00 mg·L^−1^ COD.Anode: Ruthenium- Iridium-Titanium (Ru-Ir-Ti) net.Cathode materials: copper sheet, titanium sheet, Ru-Ir-Ti net and graphite sheet.Current densities: from 35 to 95 mA/cm^2^.Temperatures: from 15 to 45 °C.Applied potential: 3.5–5.5 V.Cu^2+^ recover: up to 89.3% in 80 min; purity Cu^0^ 98.34% by ICP.Optimal conditions: Copper sheet as cathode; current density: 75 mA/cm^2^; 1.5 cm electrode distance; 15 °C.[14]Compare Cu electrodeposition vs. Cu precipitation for treating the depleted solutions obtained in the biomachining of copper pieces.WW composition: 9.0 g Fe^2+^ L^−1^; 10.0 g Cu^2+^ L^−1^; pH 1.8.Electrodes: Pt Winkler.Temperature: 24 °CConstant voltage (10 V) vs. constant current (2 A) compared.Both alternatives need previous Fe removal and allows Cu recovery (>98%) at reasonable cost. Constant voltage operation preferred with an estimated cost of 1.43 €·kg^−1^ Cu.[18]Removing Cu(II), Co(II), Ni(II) and Ag(I) from wastewater while generating high-power electric energy.Synthetic WW: CuSO_4_ (0.0005–0.3 M); CoSO_4_ (0.01–0.1 M); NiSO_4_ (0.01–0.1 M; (NH_4_)_2_SO_4_ 1 M.Electrodes. Cathode: copper; Anode: zincCathode: stirred with an egg-shaped stir barTemperature: 18–23 °C.Applied potential: 0.13–0.32 V.Cu recovery: >90% Optimal energy density of 4888 Whm^−3^
Optimal power density of 546 Wm^−2^
[19]Study and optimization of copper recovery with pulsed electrodeposition.Real WW from jewelry industry: 0.21 mg·L^−1^ silver, 0.79 mg·L^−1^ nickel, 430 mg·L^−1^ copper, <15 ppm gold and 8.3397 ± 0.66 g·L^−1^ cyanide; pH 1.74.pulsed electrodeposition (30 min)Rotating cathode, Pattern 616 A—Princeton Applied ResearchAnode: hollow cylindrical platinum mesh33.59% copper removal (with 10 ms pulse, 190 mA, 70 rpm, and 37 °C) with a deposition efficiency of 84.36% in 30 min.[15]
Recovery of electric energy, water, and metals through an autonomous electrochemical-osmotic system (EOS).

Synthetic WW from copper-laden: CuSO_4_.Electrode: Hydrophilic carbon fiberAnode: iron plateCathode: carbon fiberPolyamide thin-film composite (TFC) FO, asymmetric TFC FO membrane and seawater reverse osmosis membrane.10 mV at open circuit and variation of cell voltage from 0.2 to 0.8 V.

Maximum electric power density of 10.5 W·m^−2^ using a spontaneous Fe/Cu^2+^ galvanic cell.98.6% recovery Cu.Electric energy production of 3.3 kW·h using a 1.0 m^2^ area membrane produces 1 kg copper and 100 L of water in 122 h of operation.
[20]
Recovery of Cu with jet electrodeposition at high current density and efficiency.
Synthetic electroplating WW: 500 mg·L^−1^ Cu^2+^, 3.15 g·L^−1^ SO_4_^2−^; pH 1.Anode: RuO_2_- coated Ti tube.Cathode: pure Ti sheet.Applied voltages: −1 to 0 V.
97.4% recovery of Cu^2+^ with current efficiency of 76.3% and current density of 12 A·dm^−2^.Total energy consumption: 5.6 kW·h·kg^−1^.
[21]
Recovery of heavy metals and simultaneous removal of phenol from multicomponent acidic media.
Real WW from petrochemical refinery. 500 mg·L^−1^ Cu^2+^; 100 mg·L^−1^ Zn^2+^; 200 mg·L^−1^ Pb^2+^; 50 mg·L^−1^ Cd^2+^; 50 mg·L^−1^ phenol Anode: IrO_2_- Ta_2_O_5_ coated Ti tube.Cathode: stainless steel.Cyclone electrochemical reactor system.Applied potential: −0.44 to −0.40 V
100% Cu recovery.Noticeable recovery of Zn, Pb and Cd and removal of phenolEnergy consumption of 0.012 kW·h for a 5 L batch
[16]
Electrochemical recovery of copper and nickel from acid pickling solutions at pilot scale.
Aqueous water acid pickling solutions: Ni (6–19 g·L^−1^); Cu (2–74.1 g·L^−1^); COD (1250–40000 mg·L^−1^); H_2_O_2_ (0–5 mg·L^−1^); pH (1–6); Conductivity (20.3–232 mS·cm^−1^)Anode materials: DSA^®^, graphite and lead.Cathode materials: Stainless steel and brass.Current density: 3–30 A·dm^−2^.Temperature: 25–50 °C.Applied potential: 0.1–0.89 V.
Optimal removal of Cu: 100% laboratory; 40% pilot. Very low Ni removal (<20%).Optimum energy consumption: 2 kWh·kg^−1^ Cu.Promising results for Cu recovery from cost analysis.
[23]
Copper recovery from treated wood waste by sulfuric acid extraction and electrodeposition.

Real WW from wood waste: copper quaternary (1890 mg·kg^−1^ Cu) and copper azole (1890 mg·kg^−1^ Cu)Intensities from 1 to 10 A.Anode: titanium coated with iridium oxide (Ti/IrO_2_).Cathode: stainless steel or copper.

92% Cu deposition at 10 A for 90 min.65 US$·tn^−1^ of profit for wood waste treatment using electrodeposition according to an economic analysis.
[17]
Tellurium recovery from spent Te electrolytes by cyclone electrowinning.

Composition of Te electrolytes: 40.02, 4.96, 4.64 and 0.52 g/L of Te, Sn, S and As, respectively and 39.00, 8.00, <1, <1 and 132.91 mg/L of Se, Mg, Pb, Cu and NaOH, respectively.Anode: IrO_2_–Ta_2_O_5_-coated titanium.Cathode: Titanium (Ti) sheet or stainless steel sheet (316 L SS).Applied potential: 1.8–2.15 V.

82.89% of maximum.99.90% purity of Te deposits, 95.61% of current efficiency and 1810.58 kWh·tn^−1^ energy consumption.
[24]
To recover Co and Ni through cementation (electrochemical deposition) with Al powder as electron donor.

Synthetic sulphate and chloride solutions: 1 mM Co/Ni; 4 pH.Process: 40 mm of shaking amplitude and 120 min^−1^ of shaking frequency at 25 °C for 24 h of operation.

Recovery of 52–56% Co and 61–71% Ni from sulphate solutions.From chloride solutions: with 0.1 g AC dosage and adding Al a recovery of 61–70% Co and 70–99.9% Ni recovery from chloride solutions.
[22]
To recover palladium from spent heterogenous catalyst from petrochemical industries.

Acid solution (HCl/H_2_O_2_) with heterogeneous catalyst LD-265 (Pd/Al_2_O_3_). Three-electrode glass electrochemical cell.Electrodes: flat plate graphite. 
99.07% Pd recovery with a purity of 94.02%
[25]
To recover copper by electro-electrodialysis (EED) from ammonia solutions.

Cu^+2^ concentrations from 0.01 to 0.1 mol·dm^−3^. Current density from 200 to 500 A·m^−2^.Anode: titanium mesh coated with ruthenium oxide.Cathode: stainless steel.Applied potential: 2–4 V.

Almost 80% current efficiency in Cu recovery.Ammonia complexing agent can be reused.
[26]
To obtain high-purity copper deposits from complex mixtures by electrodeposition with a centrifuge electrode.
Synthetic WW: 2–9 g·L^−1^ Cu^2+^, 2–9 g·L^−1^ Ni^2+^, 5–40 g·L^−1^ H_2_SO_4_ and 1.5 g·L^−1^ sodium dodecylsulfonate (SDS).Bipolar centrifuge Ti electrode. Cathode face (Ti); anode face (IrO_2_-Ta_2_O_5_)Room temperature.Potential: 0.8–1.0 V.
99.9% copper purity obtained.The rest of metals can be also separated.
[27]
To recover uranium from aqueous solution into magnetite (Fe_3_O_4_) formed by iron anode dissolution and electrodeposition.
Synthetic WW: 0.5 to 10 mg·L^−1^ U; pH 1.6–10.Adsorbent: Fe_3_O_4_ formed at the cathode (graphite) from iron (anode) dissolution. Sorption capacity: 53.6 mg·g^−1^.Room temperature.Electrode gap from 4 to 19 cm. Potential: 10–30 V.
Maximum removal of U (88%) at pH: 2.6, 30 V, 8–10 cm electrode gap.
[28]
To recover gold from cyanide solutions using a static batch electrochemical reactor operating in an electrogenerative mode.
Cathode: three-dimensional cathodes: porous graphite and reticulated vitreous carbon (RVC) and two-dimensional cathode materials: copper and stainless steel plates).Anode: zinc.Synthetic WW: KAu(CN)_2_ with 500 mg·L^−1^ Au in 0.041 M sodium cyanide solution.Anolyte: 0.10 M sodium cyanide solution.Cathode potential vs. SHE: −0.1–(−0.05) V.
>90% of gold can be recovered in 3 h of experiment for the cathodes studied.More than 99% gold was recovered in 1 h of operation using activated RVC.
[29]


The second way to improve the performance of electrodeposition is by proposing novel reactor concepts with the aim of maximizing mass transfer or enhancing the reuse of the treated effluents. One interesting approach is that used by Campenedo de Morais Nepel and coworkers [15], who propose a pulsed electrodeposition with a rotating electrode in order to improve the removal of copper and the quality of the deposit. An additional system was designed by Ning and coworkers [21], who proposed a jet design (jet electrodeposition) that maximizes mass transfer by a direct injection of the solution on the cathode surface or Wang et al., who propose a centrifuge electrode to enhance the recovery of copper into high-purity solids from complex matrixes [27]. With this approach, it is possible to work at very high current densities (and thus at high recovery rates) maintaining also high recovery percentages and current efficiencies. A similar objective is obtained with the design of cyclone cells, that promote high flow rate of electrolyte on the cathode surface in order to enhance mass transfer of the process [24]. An additional interesting approach is the so-called electro-electrodialysis (EED), that combines the proper use of ion exchange membranes inside an electrochemical cell in order to allow the recovery of the metal and the simultaneous reuse of a solution of interest [26]. Finally, Lu and coworkers proposed a system that promotes the adsorption of uranium on magnetite (Fe_3_O_4_), that is produced on a graphite cathode from the Fe(II) dissolved from an iron anode [28]. Specific details about the metal recovery percentages and efficiencies can be consulted in Table 1.

Additionally, a very recent approach to upgrade the potential uses of electrodeposition for metal recovery is coupling this process with the generation of energy. This is the case of the work of Wang et al. [19], who propose to apply the process called bimetallic thermally regenerative electro-deposition battery (B-TREB), which uses waste heat to regenerate an ammonia solution that is used as the anolyte of a spontaneous galvanic cell that produce energy by oxidation of zinc and electrodeposition of copper. The similar aim is reached by using an electrochemical-osmotic system (EOS), that take advantage of the high salinity of a copper containing wastewater to produce energy by promoting a spontaneous electrochemical reaction involving the electrodeposition of copper and the oxidation or iron. Moreover, by placing an osmosis membrane between electrodes, it is possible to produce reclaimed water in the anodic compartment [20].

## 3. Use of Ionic Liquids in Electrodeposition

Ionic liquids (ILs) can be defined as organic ionic salts that are liquid at ambient or near ambient temperature. Among its properties, it is worth noting their low or negligible volatility, high thermal and chemical stability, high ionic conductivity, high solubility, low flammability, moderate viscosity and high polarity [30]. This interesting combination of properties have attracted an exponential increasing interest of the scientific community, with a growing number or articles published and an increasing spectrum of potential applications [31]. Among these applications, the low volatility and high electrical conductivity of ionic liquids, their potential applications electrochemical processes is continuously increasing, including their use as electrolyte in batteries and fuel cells, electrode materials for batteries/supercapacitors and carbon precursors for electrode catalysts [31,32,33,34,35].

Electrodeposition is one of these potential applications of ionic liquids in electrochemistry. The use of ILs in electrodeposition mainly tends to enhance the recovery yield of heavy metals due to the high conductivity of ILs together with their wide electrochemical window, that prevent the concurrence of hydrogen evolution [36].

Thus, Table 2 resumes the most recent and relevant papers published in the field, including the main objective, the most relevant process conditions and conclusions reached from these works and focusing mainly on those works devoted to the recovery of metals.

As it can be observed, ionic liquids are mainly used to replace aqueous environments in order to enhance the recovery percentage of high value-added metals, as gold, palladium, copper or platinum and increasing current efficiency. In general terms, using a certain percentage of ILs enhances the recovery of the target metal and allows obtaining this metal as a power of a controlled size or morphology [37,38,39,40], including some works regarding the formation of metal nanotubes [41]. It is also common to observe that the metal recovery does not improve when the concentration of the IL increases above a given threshold.

The use of ILs not only improves the recovery and morphology of the target metal, but also increases the percentage of metal extracted from mineral ores or from electronic waste as waste printed circuits or used mobile phones [37,38,42]. In this line, an approach has been done when using mixtures of ILs to prepare a deep eutectic solvent mixture, which can be used to efficiently extract metals from mineral power and further recover them by electrodeposition [43].

Although the use of ILs in electrodeposition is gaining an increasing interest, some issues should still be solved, as is the case of the low thermal and chemical stability of ILs, which hinder its reusability and thus increase its cost and environmental impact, and the high viscosity of ILs, which may lead to low mass transfer coefficients and, consequently, to mass transfer limitations [44,45].

The cost of ionic liquids is also an additional key aspect to consider for its use in the recovering of metals by electrodeposition. Although there is not a comprehensive study about the cost/benefit of using ionic liquids combined with electrodeposition, some works have presented partial conclusions on this topic. Thus, Abbott et al. conclude that the high cost and viscosity of ionic liquids make them better suited for the concentration of metals from large volumes of aqueous metal solutions to reduced volumes of ionic liquids concentrated in the target metals to be recovered [36]. Additionally, Magina et al. pointed out in their recent review about the challenges and opportunities in the use of ionic liquids, that the high cost of ionic liquids is one of the main drawbacks for the use of these group of compounds [31]. In the same line, Parmentier et al. concluded that the use of tetraoctylphosphonium oleate for cobalt concentration was more expensive in both CAPEX and OPEX than a conventional process using ion exchange resins, although they pointed out that it could be a promising alternative for the recovery of precious metals or when the brine disposal is a matter of concern [46].

To sum up, although the use of ionic liquids is a promising alternative, a detailed study of the environmental and cost issues should be carefully performed for any application to be developed.


molecules-26-05525-t002_Table 2Table 2Use of ionic liquids for metal recovery.ObjectiveProcess ConditionsConclusionsRef.Evaluating the properties of N,N,N-Dimethylbutylammonium Methanesulfonate [DMBA][MS] for Pb electrodeposition from Pb-acid batteries.Synthetic Pb solutions (from 0.025 M to 0.1 M Pb) with increasing water contents (from 20 to 60%).Cathode: copper foil.Anode: lead wire.Stable electrode potential −0.510 vs. SHE.Electrochemical properties of IL strongly depend on water content and acid to base ratio. The morphology of the deposit depends on the potential.[47]Studying the deposition of Ta nanotube arrays by porous anodic alumina (PAA) assisted electrodeposition using 1-butyl-1-methylpyrrolidinium bis(trifluoro-methylsulfonyl) imide ([BMP]Tf2N as solvent.Cathodes: Sitall substrate with Ni/Cr adhesion layer and sputter-deposited gold layer on the top/ PAA template (100 nm wide/1 μm length pores).Anode: platinum wire.Potentiostatic deposition at −1.4 V at 200 °C.The main component of nanotubes was tantalum pentoxide.Synthesis of nanotubes up to 900–1000 nm long and diameters of 100 nm, characterized as semiconducting.[41]Comparing the efficiency of five new imidazolium ionic liquids for the extraction of Pt (IV) from mixed metal solutions and its further electrodeposition.Pt extraction experiments from mixed metal solutions with Zn^2+^, Fe^3+^, Cu^2+^, Ni^2+^ and Rh(III) (2 mM each).Cathode for electrodeposition: Cu film (10 × 10 mm).Potentiostatic deposition at −1.75 V.ILs successfully extract Pt from mixed metal solutions.Pure Pt coating and Pt nanoparticles with diameters of 2–10 nm were identified as the electrolytic products. [48]Improving the recovery of copper from waste printed circuit boards (WPCBs) by replacing H_2_SO_4_ by the ionic liquid [BSO_3_HPy]∙HSO_4_.WPCBs powders dried and microwave-digested: 20.6% Cu; 4.55% Zn; 4.3% Al; 3.73% Fe; 2.36% Pb; 0.42% Sn. IL: [BSO_3_HPy]·HSO_4_.Slurry electrolytic system.Cathode: titanium net.Anode: graphite rod.Working conditions: 0.5 A for 3 h.At the optimal conditions (10% H_2_SO_4_ was replaced by IL) the recovery rate, current efficiency, purity and particle size of copper powders were 90.94%, 70.68%, 81.69% and 2.30 μm, respectively. [37]Enhancing the total metal recovery from waste printed circuit boards (WPCBs) by replacing H_2_SO_4_ by the ionic liquid [MIm] HSO4.WPCB metal-enriched scraps dried and microwave-digested: 71.96% Cu; 2.92% Pb; 2.43% Fe; 1.22% Al. IL: [MIm]HSO_4_.Slurry electrolysis system. Cathode: copper plate.Anode: ruthenium-plated titanium plate. Working conditions: 1.624 A (80 mA·cm^−2^), for 4 h. H_2_SO_4_ substitution by IL increase recovery rate of Al, Fe, Sn and Zn but decreases that of Cu, Mg, Ni, and Pb. For 80% H_2_SO_4_ replaced: total metal recovery: 85%; purity: 89%; current efficiency: 52%; particle size of cathode metal powder; 3.77 μm.[38]To study copper leaching from waste mobile phones with different ILs and evaluate the influence of several operational variables.Waste printed circuit boards (PCBs): 83.31% Cu; 3.35% Fe; 12.44% Zn.ILs: [Bmim]Cl; [Emim]Cl; [Bmin][BF4]; [Bmim][PF6].Slurry electrolysis system.Cathode: titanium plate.Anode: carbon rod.[Emim]Cl and [Bmim][PF6] are the best ILs for Cu leaching.Copper leaching efficiency was improved with the addition of H_2_O_2_. Maximal Cu recovery of 92.65% with a current efficiency of 79.99%.[42]To improve the permeability of gold from aurocyanide solutions with an ionic liquid-based polymer inclusion membrane (PIM) technology integrated with an electrodeposition unit.[A_336_][SCN] as the extractant with PVDF as support.ED separated from the feed solution (50 ppm Au^+^) by a PIM.Cathode: copper.Anode: graphite.98.6% Au transported and 96.4% deposited.A constant voltage (1.5 V) applied to the stripping solution enhances Au permeability.The PIM shows outstanding stability.[49]Extraction and recovery of neodymium from acidic medium using extraction by novel undiluted and non-fluorinated ILs and electrodeposition.Nd acid solutions with concentrations between 5·× 10^−4^ and 5 ×·10^−2^ M.ILs: 1-octyl-1-methylpiperidinium octylphos-phitePip18-OP, 1-octyl-1 methylmorpholiniumoctylphosphiteMo r18-OP and 1-octyl-1-methylpyrrolidinium octylphosphitePyr18- OP.Potentiostatic electrolysis at −2 V.Cathode: copper disk electrode.Anode: neodymium rod.Current efficiency higher than 83%.48% purity in Nd deposits: crystallite rods of 3–70 µm in length and 0.5–30 µm diameter.[39]Extract gold from ore in b_mim_HS0_4_.Ore: Si0_2_ (81%*w*/*w*), CaS0_4_ (10%*w*/*w*), Fe_2_0_3_ (6%*w*/*w*), Al_2_0_3_ (2.5%*w*/*w*), Mn0_2_ (0.3% *w*/*w*) and Ti0_2_ (0.2%*w*/*w*); 42 g·tonne^−1^ Au.IL: 1 -butyl-3-methylimidazolium hydrogen sulfate (b_mim_HS0_4_).Current density: 5–200 mA·dm^−2^.Cathode and anode: platinum.Gold extraction in the presence of thiourea is high yielding (86%).Current efficiency of 30% for the 20% b_mim_HS0_4_ solution at a current density of 5 mA·dm^−2^.[50]Use ionic liquids as deep eutectic solvents (DES), for the recovery of different metals by electrodeposition.Dissolution of a mineral powder (multielemental: Cu, Fe, Pb, Ni, Zn, Al, Au, Co, Mn, Ag and Cr).IL: Choline chloride, malonic acid and ethylene glycol were used to make the hybrid DESs.Electrode: titanium.Anode: platinum.With a small concentration of DES added, the recovery of Cu, Fe, Pb, Ni, Zn, Al, Au, Co and Mn increased. Ag and Cr recovery increases with a high quantity of DES.[43]Removal of iron and boron by ionic liquid extraction and recovery of neodymium by ED from voice coil motors (VCMs) waste.Nd-Fe-B magnets from VCMs.ILs: phosphonium: tri-n-butyl phosphate (TBP) and triethylpentylphosphonium bis(trifluoromethylsulfonyl)amide ([P_2225_][TFSA])Cathode: Cu substrate.Anode: prismatic Nd-Fe-B rod.%Recovery yield Fe: 95,1% with −1.75 V applied.%Recovery yield Nd: 75% with −3.25 V applied 100% B removed.[51]To study gold(I) recovery from alkaline cyanide solutions by using hydrophobic imidazolium-based ionic liquids as extractants.ILs: 1-butyl-3-methylimidazolium hexafluorophosphate ([C4_mim_][PF6]), 1-hexyl-3-methylimidazolium hexafluorophosphate ([C6_mim_][PF6]), 1-octyl-3-methylimidazolium hexafluorophosphate ([C8_mim_][PF6]), and 1-octyl-3-methyl- imidazolium bis(trifluoromethylsulfonyl)imide ([C8_mim_][Tf2N]).Cathode: copper. Anode: platinum foil. Potentiostatic electrodeposition at −1.2 V.Recovery of 92% with 16 h electrolysis.Excellent stability of the IL[52]Recover palladium, ruthenium and rhodium by electrodeposition controlling the morphology and composition of the deposits.IL: 1-butyl-3-methylimidazolium chloride (b_mim_Cl).Electrodeposition of platinoids from b_mim_Cl.Cathode: Stainless steel.Recovery Pd: 51% with −1.2 V applied from b_mim_Cl.Recovery Rh: 26.6% with −1.7 V applied from b_mim_Cl.Recovery Ru: 10,1% with −1.0 V applied from b_mim_PF6.[40]


## 4. Use of Bio-Electrochemical Processes

The bio-electrochemical process is defined as a system where the electrochemical reactions are taking place with the contribution of a living system. In most of the cases, the living systems are based on microorganisms, but also plants and higher organisms can be used. In the bio-electrochemical processes, the living organisms mainly contribute to the oxidation processes but can also contribute to the reduction processes.

Depending on reaction spontaneity of the bio-electrochemical systems, these systems can be defined as Microbial Fuel Cells (MFCs) or Microbial Electrolysis Cells (MECs). On the one hand, MFCs carry out spontaneous oxidative and reductive half reactions, therefore exerting a net energy flow. From the oxidative and the reductive processes, chemicals can be oxidized and reduced, leading to the removal of pollutants, the recovery of precious chemical, such as metals, etc. On the other hand, MECs carry out non-spontaneous oxidative and reductive half reactions, being necessary and an energy supply to carry out the reactions and causing, therefore, a net energy consumption [53,54].

When using bio-electrochemical systems for metal recovery, single, dual or multiple electrochemical deposition cells can be used. The single configuration is the simplest. In this configuration, the air acts as the final electron acceptor, its availability on the cathode being necessary [55]. In the dual configuration, the anode and cathode are electrically connected by an external conductor, whereas both compartments are separated by a membrane [55,56,57,58,59,60]. Finally, the multiple configuration requires the participation of several units electrically connected in series or parallel and hydraulically connected individual, cascade, etc. With the multiple-cells configuration, a higher energy production and chemical transformations can be obtained. Nevertheless, it has multiple electrical and hydraulic connections, so this can hinder the assembly [55,61,62].

In the literature, during the last years different new models or configurations have been developed with the aim to improve the performance of the bio-electrochemical systems. These modifications are very important because, depending on the shape (cylindrical, rectangular, etc.), the size, the superficial characteristics, and many other parameters, the mass transfer can be enhanced, leading to an increase in the performance yields.

In Table 3, the most relevant publications related to the metal recovery by using bio-electrochemical processes are presented. In these publications, the main operational parameters and performance of the bioelectrochemical systems have been studied.

Nowadays, the bio-electrochemical systems seem to be a very interesting option of metal recovery from effluents. This is because these systems allow us to reach simultaneously two objectives: on the one hand, the metal recovery and on the other hand, the energy generation. The bioelectrochemical processes can be coupled with other technologies or processes, like coupling with a thermoelectric generation [63] or electrocoagulation to increase the pH and remove iron [63,68]. With the last, more than 94 and 99% Fe was recovered by electrocoagulation and MFC, respectively. Both obtained great results, but the MFC configuration yielded better results [68].

In the scientific studies, different substrates were fed to the microbial culture of the anodic chamber, those presenting higher yield in terms of electricity generation and metal recovery being the most biodegradable [65,73]. In other cases, real wastewaters have been coupled with the metals recovery in both synthetic and real AMD effluents [59] obtaining high recovery yields of the metals presenting the highest reduction potentials, mainly Cd, Cu, Fe, Al, Zn and Pb [64,66]. The coupling between anode and cathode has also been implemented with the objective of the antibiotic removal from hospital wastewater saving 478.88 Wh·m^−3^ of energy [67].

When using electrically enhanced systems, microbial electrolysis cells, higher metal recovery yields were obtained. These systems have the advantage that they can be done in situ [71]. Novel flow-by fixed bed bio-electrochemical reactors are also under development, allowing us to reach higher metal recoveries and energy generation efficiencies [69,70].

In conclusion, to obtain better removals and recoveries of metal is necessary to continue studying the initial concentration, molar ratios of elements from wastewaters, pH of catholyte, applied cell potential, flow regimes, inter alia as it can be proved with the scientific articles mentioned above [62,63,64,65,66,67,68,69,70,71,72].

## 5. Integration of Electrodeposition with Other Processes

The last approach that will be covered in this review is the combination of electrodeposition with other technologies. In general terms, the main aim of combining other technologies with electrodeposition is either transferring the metal from a solid to a liquid phase or improving the process of metal recovery by a previous stage of concentration/purification. Table 4 shows a selection of recent combination of electrodeposition with other technologies for metal recovery.

As it can be observed, the use of leaching or bioleaching to extract the metal from a solid to a liquid phase and thus allowing its recovery by electrodeposition is a common approach. Bioleaching consists in the extraction of metals from their ores using living organisms [74]. This technology has been extensively studied in order to enhance the metal recovery from many different solid matrixes polluted with metals [75,76,77,78,79,80,81]. Therefore, the coupling of bioleaching with electrodeposition has been extensively evaluated to recover copper from printed circuit boards, obtaining very high copper purities in the final deposits and metal recoveries ranging from 75.8% to 92.85% [82,83,84]. Additionally, this leaching can be also performed chemically, as it is the case of the work published by Wang et al., who propose the combination of chemical leaching (by a combination of acid and H_2_O_2_ addition) to obtain ultra-pure Ag deposits from spent silver oxide button batteries [85].


molecules-26-05525-t004_Table 4Table 4Integration of electrodeposition with other processes for metal recovery.ObjectiveProcess Conditions (Regarding Electrodeposition Stage)Conclusions (Regarding Electrodeposition Stage)Ref.To extract copper from Printed Circuit Boards (PDBs) waste by using bioleaching and electrodeposition sequentially.Bioleaching solution (mg·L^−1^): Cu: 2903; Ni:17; Fe: 400; Zn: 19; Mn: 6; Al: 180; Mg: 56; Na: 17Cathode: 316 stainless steel sheet.Anode: titanium coated with mixed metal oxide (IrO_2_-Ta_2_O_5_).Current density: 10 mA·cm^−2^.Electrowinning at 2 V cell voltage on average.75.8% copper from PCBs (highest pulp density) was recovered with more than 99% purity as copper foil with a current efficiency of 80.6%. The optimum time for electrodeposition operation is 3 h.[82]To recover metals (Zn, Pb and Cu) from PCDs of cell phone chargers by bioleaching and electrodeposition.PCBs: 10.8, 68.0, and 710.9 mg·L^−1^ of Zn, Pb, and Cu, respectively.Cathode: steel plate.Anode: copper plate.Potential: 2.6–2.9 V; Intensity: 0.19–0.25 A; Time 2–3 h.92.85% recovery of Cu.The use of non-pulverized PCBs for bioleaching of metals is recommended.[83]To recover high-purity copper from PCBs by bioleaching, solvent extraction and electrodeposition.Bioleaching solution (g·L^−1^): Cu: 2.196; Fe: 2.791; Mn, Pb, Sn, Zn, Al, Ni:<0.012Cathode: stainless steel.Anode: lead–calcium–tin.Potential: 1.9–2.0 V; current density: 200–250 A·m^−2^; Time: 24 hSpecific energy consumption 1.8–1.9 kWh·kg^−1^ CuCurrent efficiency of 93%.>99.83% copper metal purity.[84]To recover silver from spent silver oxide batteries by acid leaching and electrodeposition.Raw material: spent silver oxide button batteries (SR44).Cathode and anode: glassy carbon.Potentiostatic ED: −0.6 to −0.05 V.Ultra-pure silver (>99.98% wt.) is obtained.98.5% metal recovery with 98.7% energy efficiency.[85]
Recover Ni(II) ions from real nickel plating wastewater, with a pilot-scale fixed-bed ion exchange resin and subsequent electrodeposition.
Real WW from nickel electroplating: 500 mg·L^−1^ Ni(II).Anode: RuO_2_/IrO_2_ titanium plate.Cathode: titanium plate.Current density: 50–500 A·m^−2^.Temperature: 30–70 °C.pH: 1–5. Time: 8 h.
Ni recovery: 95.6%.Current efficiency: 95.8%.Energy consumption: 25.05 kWh per ton electroplating wastewater.Total cost: ¥ 80000 per ton electroplating wastewater.
[86]
To develop a membrane capacitive deionization (MCDI) system for water desalination and metal recovery.
Real saltwater: 500 mg·L^−1^ CuCl_2_ and ZnCl_2_.Anode and cathode: activated carbon fiber attached onto the graphite sheet.Voltages: 0.4, 0.8. 1.0 and 1.2 V for desalination.
Recovery Cu^2+^: 42.8%.Adsorption capacity: 108.7 mg·g^−1^ Cu^2+^; 122.6 mg·g^−1^ Zn^2+^.Current efficiency: From 24.1% to 36.5%.Energy consumption: from 1.24 Wh·g^−1^ to 1.65 Wh·g^−1^
[87]
To recover Cu(II) from wastewater through an ion exchange process coupled with electrodeposition.
Wastewater: from 98 to 266 mg·L^−1^ Cu(II).Room temperature.Anode: platinum wire.Cathode: copper plates.Scanning potential: −0.7 to −0.5 V vs. Ag/AgCl.
Energy consumption: 0.6 kWh kg^−1^ in electrolysis.Copper purity (96.38%).Cu recovery: 65% as electrolyzed copper.
[88]
To study the recovery of Cu(II) by a combination of polymer enhanced ultrafiltration (PEUF) and electrodeposition.
Synthetic nitrate solution: 160 mg·L^−1^ Cu for PEUF stage; 800 mg·L^−1^ for electrodeposition.Rotating graphite cathode (60 × 60 mm square size; 10 mm thickness).Anode: Cylindrical porous graphite.Intensity: 0.15 A; pH: 2.
Electrodeposition is a viable via for polymer regeneration in PEUF.100% copper recovery.Polymer used in PEUF is not affected by electrodeposition.
[89]


Another process with results that are interesting for enhancing the efficiency of electrodeposition technique is ion exchange. The main aim of using an ion exchange resin is to increase the concentration of the metal to be recovered in order to increase the efficiency of the subsequent electrodeposition stage. Thus, when using an ion exchange, it works in cycles of operation (production of a treated solution)–regeneration. This latter regeneration stage is performed by adding an acid stream, and it produces a concentrated metal solution that is further treated by electrodeposition. With this approach, it is possible to recover nickel with very high current efficiency (95.6%) [86] and to recover Cu from wastewater with high purity (96.38%) and low energy consumption (0.6 kWh·kg^−1^ Cu) [88].

Further approaches have been done including the combination of membrane processes with electrodeposition. In this group, it is worth noting the combination of a membrane capacitive deionization with electrodeposition in order to simultaneously promote the desalination of a water stream with the recovery of copper, concluding that the recovery of copper is possible by the combination of technologies proposed [87]. Another example of combination of membrane technologies and electrodeposition was proposed by Camarillo et al., who combined the so-called polymer enhanced ultrafiltration technique (PEUF) with electrodeposition as an efficient process for efficient copper recovery [89]. In this latter work, PEUF uses a water-soluble polymer to recover and concentrate Cu(II) by an ultrafiltration membrane. This concentrated is further treated by electrodeposition, a technology that allows us to recover copper and to simultaneously regenerate the polymer, that can be used in a further PEUF stage.

To sum up, great recovering percentages, acceptable energy consumption and high metal purities can be obtained from both solid and liquid wastes by properly combining electrodeposition with leaching or concentration techniques.

## 6. Conclusions

This review covers the most relevant and recent uses of electrodeposition for metal recovery. From the information included in this this review, the following conclusions can be deduced:

The scientific interest in the use of electrodeposition is continuously growing since it was first reported in 1904.The use of electrodeposition in water treatment and reuse is probably the most important topic regarding the use of the technology for metal recovery, with the most recent works mainly devoted to the development of novel reactor configurations of enhanced mass transfer characteristics.When using electrodeposition combined with ionic liquids, it is possible to obtain an elevated yield of value-added metals recovery and a controlled morphology and size of the deposits. The cost, stability and reusability of ILs is a matter of improvement for the development of the technology.The attention devoted to the use of bio-electrodeposition systems has increased within the last years, as the selection of the reactor configuration, operational conditions and source of the inoculum are critical in order to obtain the best performance of these systems.Electrodeposition is commonly coupled to other technologies that allow either extracting the metals from a solid phase or concentrating them in the liquid phase, leading to an upgrade of the metal recovering while saving energy.

## Figures and Tables

**Figure 1 molecules-26-05525-f001:**
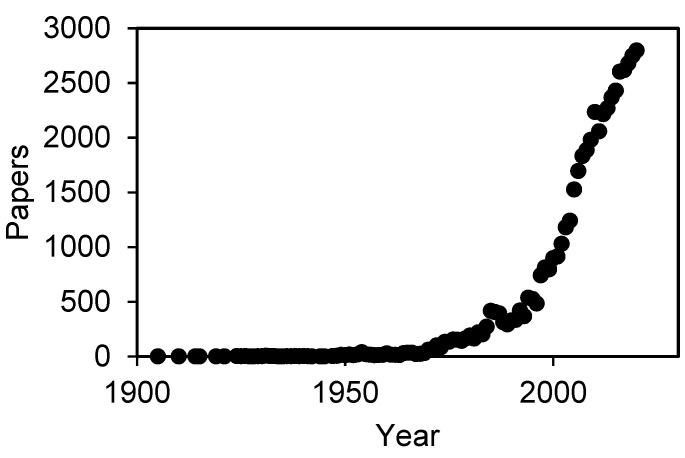
Number of papers published related to “electrodeposition” appearing in either abstract, title or keywords.

**Figure 2 molecules-26-05525-f002:**
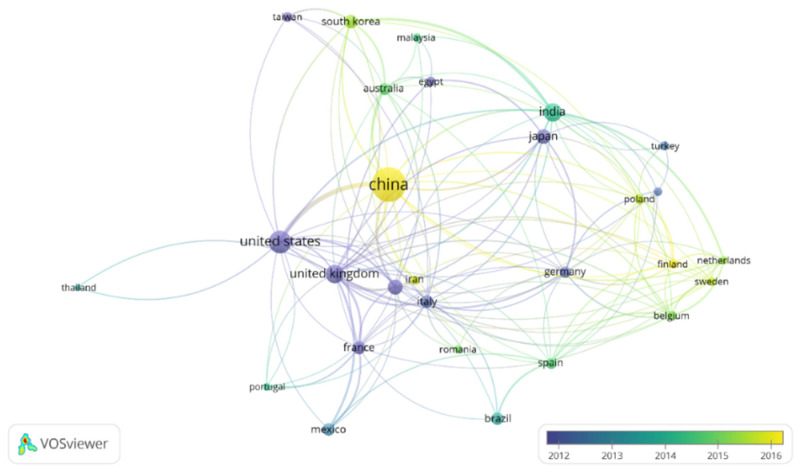
Publication density of countries by years including the term “electrodeposition” and “metal recovery”. Minimum number of publications by country: 13. Figure made with the software *VOSviewer*.

**Figure 3 molecules-26-05525-f003:**
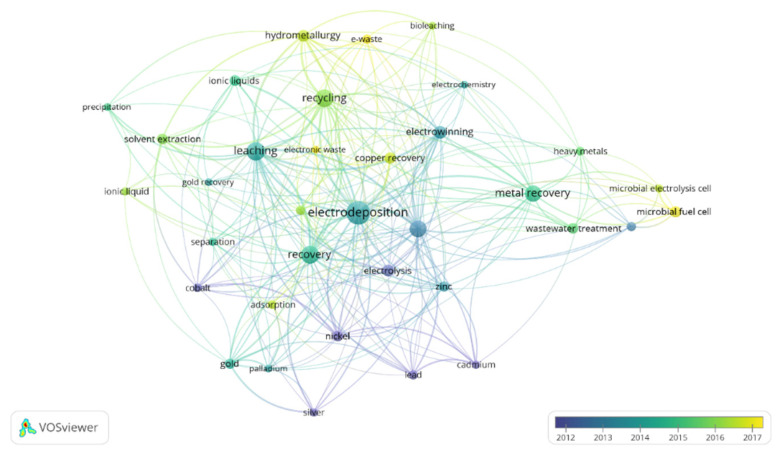
Keywords of papers published per year including the terms “electrodeposition” and “metal recovery”. Minimum number of keyword occurrence: 13. Figure made with the software *VOSviewer*.

**Table 3 molecules-26-05525-t003:** Bio-electrochemical processes for metal recovery.

Objective	Process Conditions	Conclusions	Ref.
Treating smelting wastewater by a bioelectrochemical system (BES) coupled with a thermoelectric generation (TEG), which uses the simulated heat potentially available in the smelting factories.	Cathode fed with synthetic smelting WW: 267.59 mg·L^−1^ Cu^2+^; 140.88 mg·L^−1^ Cd^2+^; 130.16 mg·L^−1^ Co^2+^; pH 1.8. Anode fed with simulated organic WW. BES operated sequentially as MFC and MEC (at 1 and 2 V). BES coupled with thermoelectric generator.	Cu^2+^, Cd^2+^ and Co^2+^ in smelting wastewater were sequentially (Cu-Cd-Co) recovered at a rate of 121.17, 158.20 and 193.87 mg L^−1^ d^−1^, respectively. Cu^2+^ recovered as Cu^0^, Cd^2+^ recovered as Cd(OH)_2_ and CdCO_3_, Co^2+^ recovered as Co(OH)_2_. COD removal of 60.55 ± 1.22%.	[63]
Co-treatment of municipal wastewater (anode) and acid mine drainage (cathode) by dual-chamber microbial fuel cells (DC-MFC).	Anode fed with a 50% municipal WW and 50% sewage sludge. Cathode fed with an industrial acid mine drainage: 42.53 mg L^−1^ Fe; 412.38 mg L^−1^ Zn; 51.50 mg L^−1^ Al; 248.00 mg L^−1^ Mg; 2.91 mg L^−1^ Cd; 1.34 mg L^−1^ Cu; 2.90 mg L^−1^ As; 36.48 mg L^−1^ Mn; pH 2.50. Two cells operated, one with a 100 Ω external resistance and the other at open circuit potential. Electrodes: graphite rods. MFC voltage: 100–500 mV.	Both effluents simultaneously treated by the DC-MFC system 15% COD removal from municipal WW and 42, 84, 71, 77 and 55% removal of Cd, Cu, Fe, Al and Pb, respectively. Maximum volumetric power of 14,000 mW m^−3^ with a MFC voltage of 300 mV.	[64]
Observe the results in the start-up, maximum voltage output, power density, coulombic efficiency and microbial communities due to the different energy substrates used.	Synthetic AMD 348.87 mg·L^−1^ Cu^2+^, 45.06 mg·L^−1^ Fe^3+^, and 7.03 mg·L^−1^ Fe^2+^; pH 1.8. MFC reactor (28 mL); resistance of 1000 Ω Anode: Carbon brush has the biofilm. Cathode: Carbon cloth treats the AMD. Membrane: Anion exchange membrane Anodic substrates: glucose, acetate, ethanol and lactate. Voltage output: 0–212 mV.	Acetate-fed-MFC showed the highest power density (195.07 mW·m^−2^). All Cu^2+^ ion (349 mg·L^−1^) was reduced as Cu^0^ in 53 h. 100% recover Cu^2+^ with lactate and ethanol. >90% recover Fe with abiotic, glucose, acetate, ethanol and lactate.	[65]
Elaborate a bioelectrochemically assisted electrodeposition system for the removal and recovery of Pb and Zn.	Synthetic smelting WW: 115 mg L^−1^ Pb^2+^, 150 mg L^−1^ Zn^2+^, and small amounts of Cd^2+^ and Cu^2+^ (less than 1 mg L^−1^ each); pH 5. Bioelectrochemically assisted electrodeposition system. Anode: carbon brush; 200 mL; inoculated with sludge fed with MWW (NaAc, 0.2; NH_4_Cl, 0.15; NaCl, 0.5; MgSO_4_, 0.015; CaCl_2_, 0.02; NaHCO_3_, 0.1; KH_2_PO_4_, 0.53; K_2_HPO_4_, 1.07; 1 mL trace element per liter). Catholyte: graphite rod; 200 mL simulated smelting WW. Membrane: anion exchange membrane (AEM). Temperature: 25 °C.	Electrodeposition equilibrium constant k_0_ = 3.76 cm·s^−1^. Zn recovery: 98.7 ± 0.7% in 6 h with −1.2 V. Pb recovery: 98.5 ± 1.4% in 10 h with −0.75 V vs. Ag/AgCl. COD removal efficiency in the anode of 76.6% at 20 h.	[66]
To demonstrate the feasibility of chloramphenicol (CAP) removal and Ag recovery.	WW: 5 mS·cm^−1^ conductivity. In MFC 300, 400 and 500 mg·L^−1^ Ag (I); pH 1 and in MEC 20, 25 and 30 mg·L^−1^ CAP. MFC (to recover Ag) and MEC (to convert CAP; copper foam as cathode electrode). Electrode: several graphite fiber connected by titanium wire. Anode and cathode: graphitic carbon rod. Cation exchange membrane. Room temperature and pH maintained: 2. Applied voltage to MEC: 0.1–0.9 V.	The optimum condition was 99.8% of Ag(I) in MFC and 98.8% of CAP in MEC. Process of 15 days. One control experiment was also carried out. The system was also operated under open circuit conditions (OCCs). 478.88 W·h·m^−3^ energy saved.	[67]
To compare the efficiency of microbial fuel cells (MFCs) with dual chamber and electrocoagulation (EC) for pH increase and Fe removal.	Synthetic Fe-rich AMD: 500 mg/L Fe; 2.4–2.5 pH. Both in batch reactors, Bioanode inoculated from municipal WW, AMD as cathode, anion exchange membrane (AMI7001), at room temperature, initial pH catholyte at 2.4–2.5. EC testing: electrode materials (Fe and Al), reaction times (15–90 min), current intensities (100–500 mA) and at 6 V. Total volume of 500 mL Rotational speed of 200 rpm. Electrode: materials (Fe and Al), plates. Final pH catholyte: 7.3–7.4. MFC at open and closed circuit: total volume of 650 mL. Electrode: Graphite bars with a smooth surface. Final pH catholyte: 7.7–7.8. Initial pH anolyte: 6.3–7.1. Final pH anolyte: 6.2–7.5.	EC: The Fe recovery of >94% was at 500 mA and 60 min. EC in its optimal conditions (Fe-electrodes, 400 mA and 60 min) was found to be more cost effective than MFC and more suitability for large-scale use. EC energy consumption is 9.6 and 3.6 kWhm^−3^ for Fe and Al, respectively. Operating cost for EC is 0.56 and 0.81 $·m^−3^ for Fe and Al, respectively. MFC: Power density between 2 and 20 W·m^−3^. The Fe removal of >99% was achieved thanks to a 1000 mg·L^−1^ acetate at the beginning. Total cost of consumables for MFC 2731 $·m^−3^.	[68]
Study applied cell potential, initial cobalt concentration, pH of the catholyte, and the mesh size of the cathode on the performance of a new design of flow-by fixed bed bio-electrochemical reactor.	Synthetic WW: CoCl_2_ solution adjusting the pH by addition of NaOH or HCl. Anolyte: prepared solution with soil. Anode: rectangular porous UHP graphite with 20–26% porosity. Cathode: stainless steel screen (316-AISI) with mesh size 30, 40, and 60 in^−1^. Cation exchange membrane (IONIC-64 LMR). Temperature: 25 ± 2 °C. Initial concentration studied: 25; 50; 75; 100 and 125 ppm. pH studied: 1; 3; 5; 7 and 9. Voltages studied: 0.6; 0.9; 1.2; 1.5 and 1.8 V.	The best removal efficiency of cobalt is >99% with 1.8 V; 50 ppm initial concentration; pH of 7; a tack of stainless no. 30 steel mesh as a packed bed cathode; energy consumption 1564 kWh·kg^−1^ cobalt.	[69]
Use of polyaniline coated electrodes to recover cadmium in semiconductor processing WW.	WW from semiconductor processing from a middle process: 400 mg L^−1^ Cd ions; 85 mg L^−1^ COD; pH 5.8. Electrodes: Polyaniline coated on carbon-base material (PAC). Temperature: 30–55 °C. Initial pH: 7. Voltage: 5–20 V. Cadmium concentration: 200–800 mg L^−1^.	92% turbidity removal, 89% of TSS; 90% Cd recovery and 72% of water reclaimed in 40 min. Polyaniline coated electrodes exhibit high capability for the electrochemical recovery of cadmium.	[59]
Study the effect on copper electrowinning of the cyanide/copper molar ratio, the potential applied to the electrolytic cell, the temperature and circulation rate of the solution, and the use of synthetic solutions using electrolytic reactors in discontinuous, discontinuous with recirculation, and continuous flow regimes.	Synthetic WW from mining company: [CN]/[Cu] 3, 4 and 5 molar ratios; pH 10.9 and 6.8 mS·cm^−1^. Copper electrowinning (1 h) in discontinuous regime (stirred at 600 rpm) and Copper electrowinning in discontinuous regime with recirculation (stirred by recirculating): electrode spacing of 3 cm, graphite anode, 304 stainless steel cathode. Copper electrowinning in continuous regime with recirculation: three plate electrodes, graphite anode, two stainless steel cathodes. Applied voltages: 3; 4; 5 and 6 V. Temperatures: 25; 40; 60 °C.	The best option, according to them, is 50% copper removal applying 4 V. However, they achieve 79% Cu removal with 4 V, [CN]/ [Cu] of 3, an 1 h of electrowinning and 25 °C. Additionally, 99,90% Cu removal with 6 V, 60 °C, 1 h of electrowinning and recirculation rate of 160 cm·min^−1^.	[70]
Recovery of copper in situ in an enhanced microbial copper recovery cell with varying voltages, run it in open circuit mode and doing one abiotic	Synthetic WW from low-strength copper-laden water (0.4 g·L^−1^ sodium acetate, 4.28 g·L^−1^ Na_2_HPO_4_, 2.45 g·L^−1^ NaH_2_PO_4_·H_2_O, 0.31 g·L^−1^ NH_4_Cl, 0.13 g·L^−1^ KCl, 5 mL vitamins and 12.5 mL minerals). Electrically enhanced microbial copper recovery cell (MCRC). Anolyte: Inoculated with domestic wastewater. Anode: Heat-treated carbon fiber brushes. Cathode: Graphite sheets. Voltages studied: 0; 0.4; 0.8 V. Anion and cation exchange membrane. Temperature: 23 ± 3 °C.	Maximum current density: 1.62 ± 0.01 A·m^−2^. The maximum copper removal efficiency of 95.4 ± 0.9% was obtained thanks to maximum current density, initial concentration of copper of 5 mg·L^−1^ and 0.8 V applied.	[71]
Recover copper with two control strategies: maximum power point tracking (MPPT) and the application of 0.5 V doing 3 experiments with different WW.	Real and synthetic WW from distillery effluent: 20 L copper sulphate varying the concentration (1000 and 50 mg·L^−1^) and real WW, 1.5 g·L^−1^ NaCl; pH 4 and 2.5 mS·cm^−1^ conductivity. Two hydraulically cells connected. Anode: Four previously enriched helical. Cathode: copper foil. Anaerobic cathodes separated by polypropylene separator. Tube anion exchange membrane AEM and a cation Exchange Membrane (CEM) tube. Temperature: 23–25 °C. 0.5 V applied voltage.	The recovery percentage depends on the starting concentration. Recovery of Cu 60–95%. The recovery was low when the anode was supplied with copper depleted distillery waste.	[72]
Recover platinum through charring biofilms in MFCs, studying its distribution and generating Pt/C catalyst.	Synthetic WW: 32.92 g·L^−1^ K_3_[Fe(CN)_6_], 5.46 g·L^−1^ Na_2_HPO_4_·12H_2_O and 1.22 g·L^−1^ NaH_2_PO_4_·2H_2_O; 7.0–7.2 pH. 5 rectors, one of them was the abiotic. The reactors differ from Pt initial concentration. Electrodes: graphite plate. Anolyte: inoculated with the anaerobic sludge. Cation exchange membrane (CEM, 13 cm^2^, CMI7000, Dupont, USA). Temperature: 25 ± 1 °C. Voltage: 0–0.8 V.	Around 40% Pt was recovered. Each batch process worked for 24 h. Maximum power density of 844.0 mW·m^−2^, having 2.11 mg·L^−1^ Pt(IV).	[73]

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
