# Peer review of "An Old Technique with A Promising Future: Recent Advances in the Use of Electrodeposition for Metal Recovery"

_molecules, 2021, doi:10.3390/molecules26185525_

Round 1
Reviewer 1 Report
All work performed is based on the electrodeposition technique.
The authors were very precise throughout the text presented and still discuss several variations in the use of electrodeposition by engineering.
There is a review of literature data and in-depth discussion on this important topic.
Finally, the text is interesting for publication and therefore necessary.
Author Response
The authors want to thank the positive comments of the reviewer.
Reviewer 2 Report
This paper collected the recent research works on the recovery of metal via electrodeposition. Advances are classified in three categories; treatment and reuse of wastewater by electrodeposition, use of ionic liquids in electrodeposition, and use of bio-electrochemical processes. Recovery of metals has become a important issue and this spaper certanly provides certain information for this goal. I think it can be published in this journal. However, it would be helpful if some schemes or figures are extracted from the reference papers and included in this paper to illustrate the ideal of using electrodeposition for metal recovery.
Author Response
As in the case of Reviewer 1, we also thank the positive comments of this reviewer.
Regarding the suggestion of including additional figures, we have applied for permission to Elsevier to use three figures in the revised version of the manuscript. Specifically, we applied for permission for the use of Figure 2 of reference 24, Figure 2 of reference 21 and Figure 10 of reference 89. These figures are examples of cell designs or processes configurations regarding the recovery of metals from electrodeposition. Unfortunately, we have not received the authorization from Elsevier for the use of these figures in the time available to perform the review, so we are unable to attend to this reviewer suggestion at this time.
Reviewer 3 Report
The authors report a rather accurate bibliography concerning the recovery of precious metals by electrodeposition.
Their subdivision is questionable but can be accepted even if their discussion does not enter into the merits of the fundamental parameters of electrodeposition. In other words, there is no discussion on the variation of the currents / potentials as a function of the electrolytes or metals to be recovered.
Do you use similar currents / potentials when you need to recover the same metal?
Does the selected current or potential depend on some general rule?
Can we come up with some scientific information or general rules for the choice of currents as a function of metals?
As regards ionic liquids, I believe that an analysis of the costs / benefits should be made given the particularity of the means used.
However, I still consider the work acceptable even if I consider a major revision necessary.
Author Response
Regarding the comment on the potentials/currents selection, as the reviewer knows, the cell voltage is the potential difference between the anode and the cathode in an electrochemical cell. In an electrolytic or driven cell, it is the driving force for the current to flow through the system. The minimum or equilibrium cell potential for the current to flow is related to the Gibbs free energy of the cell, that depends basically on the metal to be recovered.
When a current flow through the system, additional resistors are added to the equilibrium potential and the cell potential is then a function of different contributing terms. These terms include cathode and anode overpotentials, arising from the need to drive the kinetics of the charge transfer at the two electrodes [1]. Those terms dominate at low current densities and can be minimized by selecting appropriate catalyst for the desired reaction and/or operating at higher temperatures. On the other hand, concentration-polarization overpotentials become important with low reactant concentrations and/or high current densities that result in the depletion of the reactant at the electrode surfaces. Therefore, an adequate approach to minimize concentration polarization over-potentials may be the use of high-surface electrodes and high mass transport conditions. The last term is the sum of all the ohmic resistors that may be found in the system. i.e. cables, electrical connections, electrodes, current feeders, electrolyte, membrane, etc. (an additional explanation, including Equations, is included in the cover letter).
To sum up, the potential applied to the system strongly depends on both the metal to be recovered and on the design of the electrochemical cell. It is not the aim of the present work to stablish a rule for selecting an appropriate working potential as a given value would be only useful for a specific metal and cell design. Moreover, it is not easy to compare data from different articles as many works present the value of the cathode potential meanwhile in many other works only the total voltage applied to the cell is presented.
- Pletcher, D., Z.Q. Tian, and D.E. Williams, Developments in Electrochemistry: Science Inspired by Martin Fleischmann. Developments in Electrochemistry: Science Inspired by Martin Fleischmann. 2014. 1-377.
- Pletcher, D. and F.C. Walsh, Industrial Electrochemistry. 1990: Springer Netherlands.
- Trasatti, S., Electrocatalysis: understanding the success of DSA®. Electrochimica Acta, 2000. 45(15): p. 2377-2385.
Anyway, in order to attend to the reviewer’s concern and to give the readers an easy access to this valuable information, we have included the value of the working potential in all manuscripts where this value is available.
Regarding the cost/benefits of using ionic liquids, in the first version of the work, we already included a brief comment on the applicability of ionic liquids and the aspects to be improved. Additionally, in order to improve the discussion on this topic, the following information has been included in the revised version:
“Although the use of ILs in electrodeposition is gaining an increasing interest, some issues should still be solved, as it is the case of the low thermal and chemical stability of ILs, which hinders its reusability and thus increase its cost and environmental impact, and the high viscosity of ILs, which may lead to low mass transfer coefficients and, consequently, to mass transfer limitations [44], [45].
The cost of ionic liquids is also an additional key aspect to consider for its use in the recovering of metals by electrodeposition. Although there is not a comprehensive study about the cost/benefit of using ionic liquids combined with electrodeposition, some works have presented partial conclusions on this topic. Thus, Abbott et al. conclude that the high cost and viscosity of ionic liquids make them better suited for the concentration of metals from large volumes of aqueous metal solutions to reduced volumes of ionic liquids concentrated in the target metals to be recovered [36]. Additionally, Magina et al. pointed out in their recent review about the challenges and opportunities in the use of ionic liquids, that the high cost of ionic liquids is one of the main drawbacks for the use of these group of compounds [31]. In the same line, Parmentier et al. concluded that the use of tetraoctylphosphonium oleate for cobalt concentration was more expensive in both CAPEX and OPEX than a conventional process using ion exchange resins, although they pointed out that it could be a promising alternative for the recovery of precious metals or when the brine disposal is a matter of concern [46].
To sum up, although the use of ionic liquids is a promising alternative, a detailed study of the environmental and cost issues should be carefully performed for any application to be developed.”
Moreover, the conclusions related to the use of ionic liquids has been completed:
“When using electrodeposition combined with ionic liquids, it is possible to obtain an elevated yield of value-added metals recovery and a controlled morphology and size of the deposits. The cost, stability and reusability of ILs is a matter of improvement for the development of the technology.”
Round 2
Reviewer 2 Report
I think this manuscript can be accepted as is.
Reviewer 3 Report
No suggestions